# communications
## engineering

# Mapping of the mechanical response in Si/SiGe nanosheet device geometries

Conal E. Murray[1✉], Hanfei Yan[2✉], Christian Lavoie[1], Jean Jordan-Sweet[1], Ajith Pattammattel[2], Kathleen Reuter[1], Mohammad Hasanuzzaman[3], Nicholas Lanzillo[3], Robert Robison[3] & Nicolas Loubet[3]

The performance of next-generation, nanoelectronic devices relies on a precise understanding of strain within the constituent materials. However, the increased flexibility inherent to these three-dimensional device geometries necessitates direct measurement of their deformation. Here we report synchrotron x-ray diffraction-based non-destructive nanoscale mapping of Si/SiGe nanosheets for gate-all-around structures. We identified two competing mechanisms at different length scales contributing to the deformation. One is consistent with the in-plane elastic relaxation due to the Ge lattice mismatch with the surrounding Si. The second is associated with the out-of-plane layering of the Si and SiGe regions at a length scale of film thickness. Complementary mechanical modeling corroborated the qualitative aspects of the deformation profiles observed across a variety of nanosheet sample widths. However, greater deformation is observed in the SiGe layers of the nanosheets than the predicted distributions. These insights could play a role in predicting carrier mobilities of future devices.

[1] IBM T.J. Watson Research Center, Yorktown Heights, NY 10598, USA. [2] National Synchrotron Light Source II, Brookhaven National Laboratory, Upton, NY 11973, USA. [3] IBM Research, Albany, NY 12203, USA. ✉email: conal@us.ibm.com; hyan@bnl.gov

The evolution of complementary metal-oxide semi-conductor (CMOS) technologies has required substantial advances in the implementation of innovative device geometries and the application of more sophisticated processing. Through such efforts, achievements in device density and performance have allowed the introduction of technology nodes to maintain their traditional cadence. New materials, such as stressed liners and heteroepitaxial source/drain regions adjacent to the device channels, have been incorporated to improve carrier mobility over the past two decades[1,2]. In addition, the transition from a planar to three-dimensional geometry, such as fin-based, field-effect transistors (FinFETs)[3–6], was adopted in an effort to mitigate short-channel effects and maintain electrostatic control, allowing the continued scaling of device dimensions. Though FinFETs have proved successful, they face several technical challenges with respect to performance, cost, and layout beyond the 5-nm technology node[7,8]. Gate-all-around structures, such as nanosheets in which the fins are composed of horizontal, semi-conducting channel layers[9,10], are being explored as the necessary evolution in device technology to replace FinFETs. However, such advancement in device geometry places great challenges on metrology and characterization techniques, which provide critical information on design and processing elements[11]. With the increased structural complexity and greater sensitivity to variations in gate-all-around FETs, the ability to accurately determine device properties becomes a challenging task due to the reduced length scales and increased elastic compliance in such geometries[12].

Predicting the performance of nanoelectronic devices relies heavily on knowledge of the piezoresistive properties of the constituent elements. While carrier mobility is related to the stress state within the semiconducting channel region through its corresponding piezoelectric tensor[13,14], alterations to this tensor in the device inversion layer[15] require a precise mapping of the strain within the actual devices. Transport measurements alone are insufficient for this purpose as they represent a nonuniform average of the mobility across the entire device. Several methods exist for the quantification of strain within CMOS technology[16–18]. Although transmission electron microscope (TEM) techniques are typically used because they possess a lateral spatial resolution that can reach below the nanometer level, the extensive sample preparation necessary to produce electron-transparent lamellae can alter the strain distributions within the regions of interest[19–21]. In contrast, synchrotron-based X-ray micro-diffraction techniques allow for a non-destructive investigation of the deformation within the strained regions of such devices[22,23] that can reside at or below the top surface of the sample.

Nanodiffraction with sub-100-nm resolution has been reported in studies of various crystalline materials[24–27]. To date, micro/nano-diffraction implementations possess a spatial resolution often found to be insufficient in studying the structural deformation within modern nanoelectronics.

Recent advancements in nanofocusing optics have greatly improved the resolution that can be achieved in x-ray microscopy[28–31]. Probes with a spot size of approximately 10 nm have been realized at several synchrotron facilities[32,33], enabling the non-destructive characterization of strain at the nanoscale. The high flux density at the focused spot provides sufficient diffracted intensity at a spatial resolution commensurate with ultra-small structures in nanoelectronics. Here we report the measurement of the strain of Si/SiGe nanosheets designed for gate-all-around FETs, where two distinct elastic relaxation mechanisms were directly observed within the nanosheets with a spatial resolution of approximately 12 nm. A combination of boundary element method (BEM) and finite element method (FEM) modeling of these structures confirms the existence of both a long-range distribution in nanosheet deformation due to their traction-free sidewalls, and a finer scale on the order of the individual layer thicknesses due to the interplay of shear and out-of-plane stress within the SiGe sections and the adjacent Si stacks. The latter represents a more dominant mechanism in dictating the strain state within the nanosheets, where gradients in the strain at such length scales can play a critical role in predicting the resulting carrier mobility of such devices and impact variation in their performance.

## Results and discussion

**X-ray nanodiffraction measurements.** Nanosheet samples were fabricated by IBM Research at the 300 mm Nanotech facility in Albany NY. Epitaxial growth of the SiGe layers was performed on 300 mm diameter, (001)-oriented silicon wafers in a commercially available, rapid-thermal chemical vapor deposition reactor. Fitting of high-resolution x-ray diffraction measurements of the blanket layers confirmed an effective, average Ge fraction of approximately $36.7 \pm 0.4\%$ prior to lithographic patterning. The TEM image in Fig. 1 depicts the nominal, cross-sectional geometry, which is comprised of three SiGe bilayers of 9 nm thickness capped with 8 nm of Si, all deposited on a Si (001) substrate. As shown in Fig. 1a, recesses are generated within the substrate to generate pillars approximately 100 nm tall, with a silicon nitride capping layer associated with the lithographic process remaining on the top of the nanosheets. The unstrained SiGe film has an equilibrium lattice constant of 5.5067 Å. Its in-plane lattice parameter is compressed due to the pseudomorphic growth process so that it matches that of the underlying Si substrate (5.431 Å). The corresponding Poisson expansion leads to a larger out-of-plane lattice spacing of 5.564 Å in a fully strained SiGe film.

Diffraction measurements were performed at the hard X-ray nanoprobe (HXN) beamline of the National Synchrotron Light Source II (NSLS-II) at the Brookhaven National Laboratory[32,34]. A 12 keV monochromatic beam was focused by a cross pair of multilayer Laue lenses to produce a spot size of approximately 12 nm[32]. As shown in Fig. 2a, the sample intercepts the beam at the focus and at a Bragg angle that excites both Si and SiGe (004) reflections in a horizontal reflection geometry. Diffraction and fluorescence signals were collected simultaneously as the beam was scanned at specific incidence angles across the nanosheet structures. The Ge Kα fluorescence edge provided a precise determination of location on the nanosheet, so that any relative positional changes due to angular motions of the sample stage could be corrected. The pixel-array detector measures a slice in

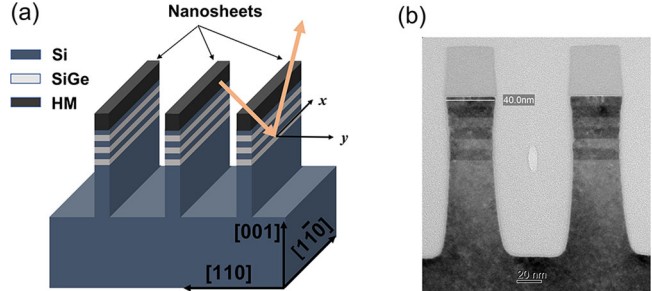

**Fig. 1 Si/SiGe nanosheet geometry. a** Schematic drawing of the sample. The nanosheets under investigation consist of a series of Si and SiGe layers lithographically patterned above Si pillars with a large length to width ratio. A silicon nitride hardmask (HM) capping layer remains on the top of the nanosheets. The light orange arrows show the incident and the diffracted x-rays. **b** Cross-sectional transmission electron microscope image of the 40 nm wide Si/SiGe nanosheets.

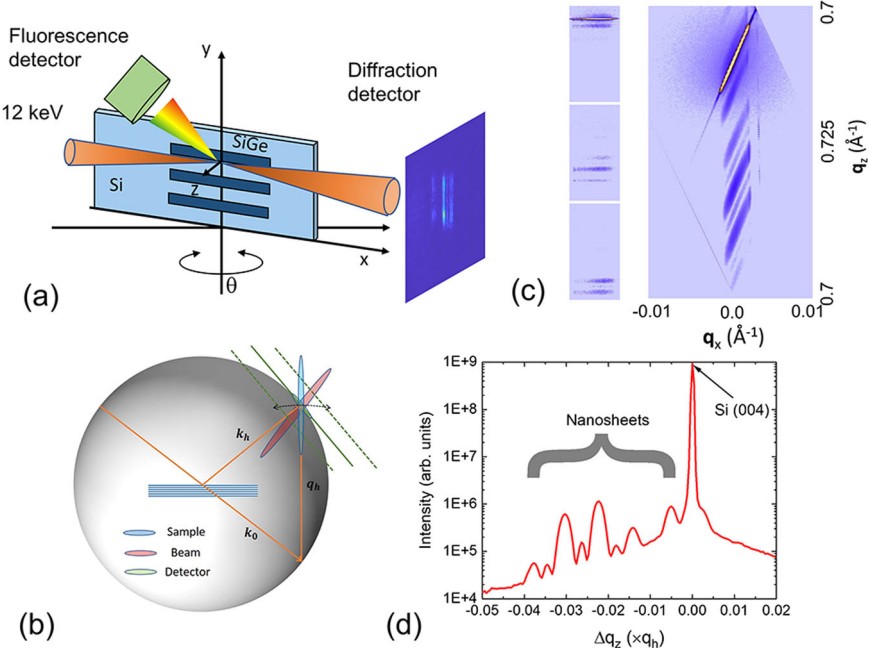

**Fig. 2 Reciprocal space mapping (RSM) of nanosheets. a** Schematic drawing of the experiment setup. **b** Depiction of the Ewald sphere, where each measurement by the pixel-array detector represents a slice of reciprocal space. As the incidence angle changes, the detector plane moves along the rocking direction of the reciprocal lattice vector, $\mathbf{q_h}$. The three colored streaks correspond to the reciprocal representations of the sample, detector, and nanobeam. **c** Recorded diffraction patterns at three incidence angles (left), and the composite 2D RSM within the crystal $xz$ plane (right), where both are plotted using a logarithmic scale. The long, tilted streaks seen in the RSM are due to the large divergence angle of the nanobeam. Note that the vertical axis $\mathbf{q_z}$ follows the reciprocal lattice vector of the (004) reflection, $\mathbf{q_h}$. **d** Radial scan of the RSM along $\mathbf{q_h}$ after integrating the intensity along the tilted streaks. Pendellosung fringes emanating from the nanosheets, as evidenced in the scan, are not impacted by the incident beam divergence.

reciprocal space cutting through the Ewald sphere, as shown in Fig. 2b. We obtain three-dimensional, reciprocal space maps (RSM) by assembling multiple slices across the reciprocal lattice vector, $\mathbf{q_h}$, acquired when rocking the sample angle. In Fig. 2c, we depict typical diffraction patterns from the nanosheets obtained at three incidence angles. Because the effective motion during data collection through reciprocal space is not perpendicular to the detector, it is necessary to apply a coordinate transformation to convert the RSM's into the crystal coordinate system, where $\mathbf{q_x}$ is parallel to this direction and $\mathbf{q_z}$ lies along the reciprocal lattice vector, $\mathbf{q_h}$. In Fig. 2c, a two-dimensional (2D) RSM is also shown, which integrates the diffracted intensity across the third dimension. We can integrate this 2D RSM along the direction of the streak to quantify any variations along $\mathbf{q_z}$ (also referred to as a radial scan), minimizing the impact due to the beam divergence. The integrated radial scan, displayed in Fig. 2d, preserves the interference (Pendellosung) fringes generated by the nanosheets despite the complex nature of the x-ray nanobeam source.

**Data analysis**. Figure 3a represents a hybrid reciprocal/real-space contour plot, which is composed of individual $\mathbf{q_z}$ profiles (out-of-plane direction), obtained from the SiGe (004) reflection as a function of lateral position near the edge of a blanket nanosheet. The shift in SiGe peak position relates directly to the change in lattice spacing due to deformation within the nanosheet, and clearly disappears beyond its edge, where the intense, Si substrate (004) intensity remains. Because diffraction from the SiGe regions contains multiple peaks, a Fourier-based method is employed to calculate the overall shift of the whole pattern at a sub-pixel resolution[35]. The variation in out-of-plane strain, $\Delta\varepsilon_{zz}$, relative to that of the fully strained, isotropic biaxial state far from the

nanosheet edge can be obtained, where the data represents depth-averaged quantities through all of the SiGe layers illuminated by the x-ray probe. In Fig. 3b we present the Ge $K\alpha$ fluorescence, $\Delta\varepsilon_{zz}$, and lattice tilt about the sample x-axis as a function of position collected from the SiGe layers in the nanosheet.

Out-of-plane deformation in the nanosheet exhibits trends consistent with elastic relaxation due to the lateral, free edges[17,25,36]: Poisson expansion, induced by in-plane compressive strain due to lattice mismatch with the underlying Si, decreases and the magnitude of (negative) $\Delta\varepsilon_{zz}$ increases. This effect can extend at least 20 times the thickness in monolithic, strained features[17], and is visible beyond approximately 300 nm from the edges of the nanosheets as seen in Fig. 3b. However, a secondary, short-range effect can clearly be observed at locations much closer to the free edge, leading to an expansion in the out-of-plane SiGe lattice parameter. The net effect of these two competing mechanisms is a maximum relaxation with a relative strain, $\Delta\varepsilon_{zz}$, of $-2.5 \times 10^{-3}$ at distance of about 40 nm away from the free edge. In the absence of the short-range enhancement in out-of-plane strain, one would expect a uniaxial stress state at the edge with a $\Delta\varepsilon_{zz}$ value of approximately $-5.6 \times 10^{-3}$.

We also investigated strain distributions within nanosheets varying in width to assess the effects due to any interaction between the two nanosheet edges. Fig. 3c, d displays contour plots of (004) radial scans and their corresponding strain profiles, respectively, as a function of position across multiple nanosheets with widths of 120, 70, 50, and 40 nm. While deformation within the 40, 50, and 70 nm wide Si/SiGe nanosheets appears to be dictated by the relaxation mechanism proximal to the edges, the profiles across the 120 nm wide nanosheets contain a central region that possesses either a flat distribution or one that exhibits a slight decrease in magnitude at the nanosheet center. It is important to note that strain distributions are not absent from the

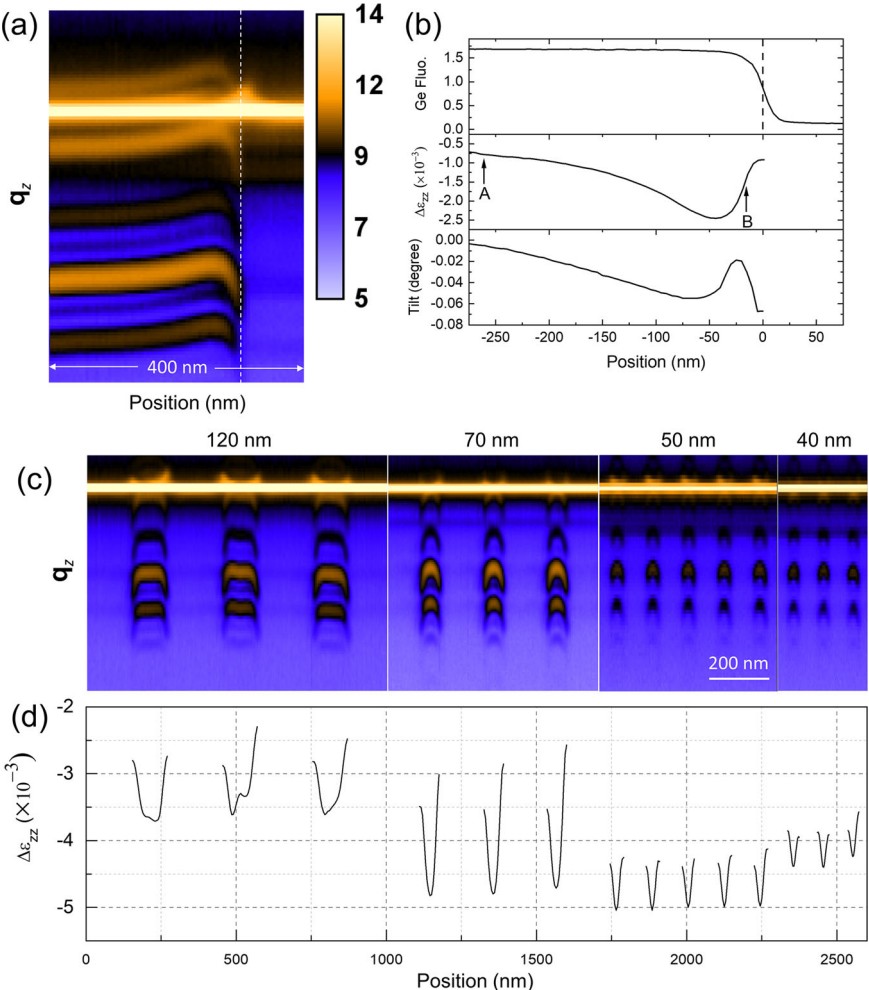

**Fig. 3 Diffraction-based measurement of nanosheet deformation. a** Contour plot of (004) diffracted intensity (on a logarithmic scale) as a function of $\mathbf{q_z}$ (vertical axis) and position (horizontal axis) across the edge of a blanket Si/SiGe nanosheet. The vertical dashed line corresponds to the nanosheet edge and the bright, horizontal intensity the substrate (004) intensity. **b** Plots of the integrated Ge Kα fluorescence (top), out-of-plane lattice deformation (middle), and lattice rotation (bottom) as a function of position near the free edge (dashed line). Two regimes of elastic relaxation are observed: reduction of Poisson expansion within approximately 300 nm from the free edge and an increase in out-of-plane SiGe deformation within 40 nm from the free edge. Lattice rotation of the SiGe layers about the x-axis of the sample represents a convolution of these two effects. **c** Contour plots of (004) diffracted intensity across the Si/SiGe nanosheet features possessing different widths. **d** Integrated plot of out-of-plane lattice deformation and lattice rotation as a function of position. In these nanosheets, the near-edge elastic relaxation mechanism dominates the observed mechanical response.

central region of the 120 nm nanosheet but rather the observed profile is a manifestation of the superposition of elastic relaxation induced by the two nanosheet edges. This mechanical response is similar to that observed in the blanket nanosheet (Fig. 3b), where it is anticipated that, beyond approximately 40 nm from the edges, the secondary relaxation effect at both edges will not interact. The resulting values of $\Delta\varepsilon_{zz}$ at the narrow nanosheet centers increases in magnitude (greater relaxation) from approximately $-3.5 \times 10^{-3}$ to $-5.0 \times 10^{-3}$ for the 120–50 nm widths due to the primary elastic relaxation mechanism. In contrast, deformation at the 40 nm wide nansosheet centers is slightly less than that of the 50 and 70 nm wide nanosheets, reversing the trend with nanosheet width.

**Depth-resolved, nanosheet strain distributions.** While the aforementioned data analysis only provides the lateral variation of depth-averaged strain, the ability to discern gradients in the strain as a function of depth within the nanosheet layers is also of great interest. By combining structural knowledge of the nanosheet geometry with a rigorous dynamical diffraction model[37], we can

retrieve the lattice spacing values in each layer by fitting the SiGe diffraction patterns. In Fig. 4a, we plot the fittings to the radial scan from the blanket Si/SiGe nanosheet stack which is under an isotropic, biaxial stress state, and the corresponding lattice spacing profile of the (004) reflection at two locations shown in Fig. 3b. The ratio of the out-of-plane lattice spacing of the SiGe layers to that of unstrained Si exhibits a value of 1.0245, which corresponds to a Ge fraction of 36.3% from Vegard's law[38] and is consistent with the expected composition. As shown in Fig. 4b, a constant lattice spacing in each layer, though different for the Si and SiGe regions, is expected in the absence of plastic deformation. However, locations near the edge of the nanosheet stack can exhibit variations in the strain field in both the out-of-plane and lateral directions due to elastic relaxation. Because a constant lattice spacing in each layer does not yield a good fit to the radial scan, we allow the strain to vary in each layer along the depth direction. When the lateral variation in strain is small over the x-ray illumination footprint (such as location A in Fig. 3b), this model works well. The poorer fit of the radial scan at location B is caused by the presence of larger, lateral strain distributions across

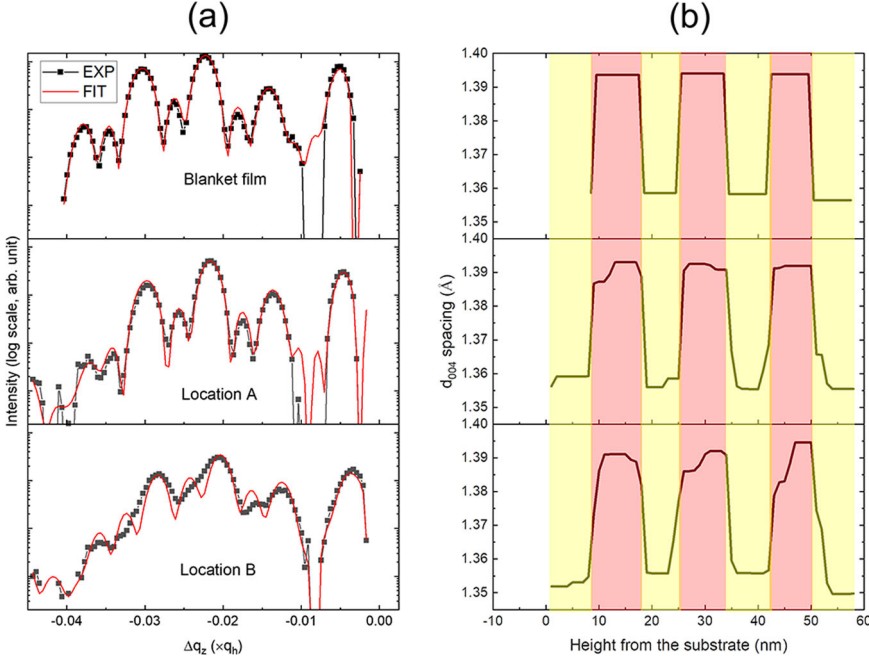

**Fig. 4 Extraction of depth-dependent deformation from site-specific diffraction. a** Fitting of the radial scans obtained from the blanket film and locations A and B near a free edge, as shown in Fig. 3b. Si substrate diffraction has been subtracted from the data. Depth profiles of the $d_{004}$ lattice spacing from these locations are depicted in **b**, where the Si and SiGe regions correspond to the larger (approximately 1.39 Å) and smaller (1.358 Å) values, respectively. Note that the red rectangles correspond to the SiGe layers, where the first one begins at a height of 9 nm, and the Si layers are shaded yellow. At locations A and B, a substrate recess of 9 nm is assumed to match the nanosheet structures (see Fig. 1b).

the footprint of the x-ray beam. While this convolution issue cannot be removed from the model, a more gradual change in the lattice spacing is observed across the Si/SiGe interfaces as one approaches the free edge of the nanosheet. We expect the secondary relaxation mechanism to induce considerable variation in the lattice deformation within each layer in the immediate vicinity of the nanosheet edge.

A more advanced deconvolution technique, Bragg ptychography, has been developed in recent years to obtain 3D strain mapping at the nanoscale[24,39,40]. However, a requirement of this technique is that sufficient coherent flux be delivered to the sample to achieve a high signal-to-noise ratio in the diffraction patterns[41]. The extremely small volumes associated with nanostructures present a major challenge in applying this technique (e.g., the maximum diffracted intensity from the nanosheet in this study is about 35 photons s$^{-1}$ pixel$^{-1}$ on the detector, inadequate for such a reconstruction). With the advent of diffraction-limited, fourth-generation synchrotron facilities[42], the increase in coherent flux may allow for the 3D strain mapping of nanostructures via Bragg ptychography reconstruction in the near future.

**Comparison to mechanical simulations**. To better understand the interplay between these two relaxation mechanisms, we employed mechanical modeling of the elastic relaxation within the Si/SiGe nanosheets. Simulations based on 2D BEM modeling[23] were conducted to visualize deformation within the nanosheet geometries for a variety of widths. As shown in Fig. 5a, reduction of the out-of-plane SiGe deformation over a length scale of approximately 500 nm as one approaches the free vertical edges of the nanosheet corresponds to the overall elastic relaxation of the nanosheet. However, an increase in out-of-plane deformation is also observed close to these free edges, consistent with the diffraction measurements. Figure 5b, which depicts the deformation in a 50 nm wide nanosheet sample, illustrates this

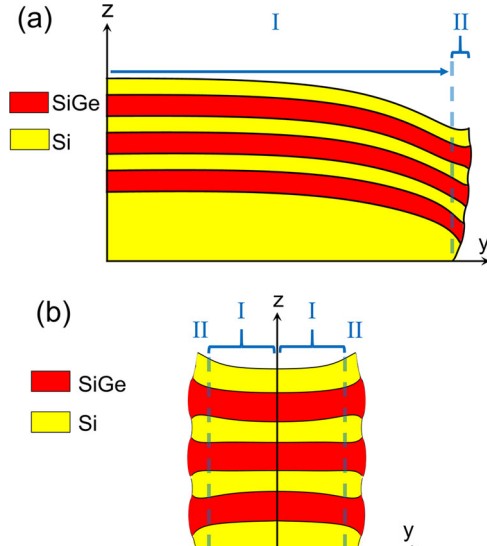

**Fig. 5 Mechanical modeling of nanosheet deformation.** Cross-sectional schematic of the amplified deformation induced **a** near the edge of a 1000 nm wide and **b** 50 nm wide Si/SiGe nanosheets as predicted by boundary element method (BEM) mechanical modeling. Two regimes of elastic relaxation are separated by the vertical dotted lines: region I corresponds to the lateral elastic relaxation due to the presence of the traction-free nanosheet sidewalls, and region II to the vertical load sharing between the Si and SiGe regions.

vertical fanning of the SiGe regions and thinning of the Si layers at the edges induced by shear strains[10] and rotations of the nanosheet lattice.

Given the fine scale of the residual strain distributions and their potential sensitivity to the processing steps associated with the nanosheets, FEM-based simulations that incorporate the

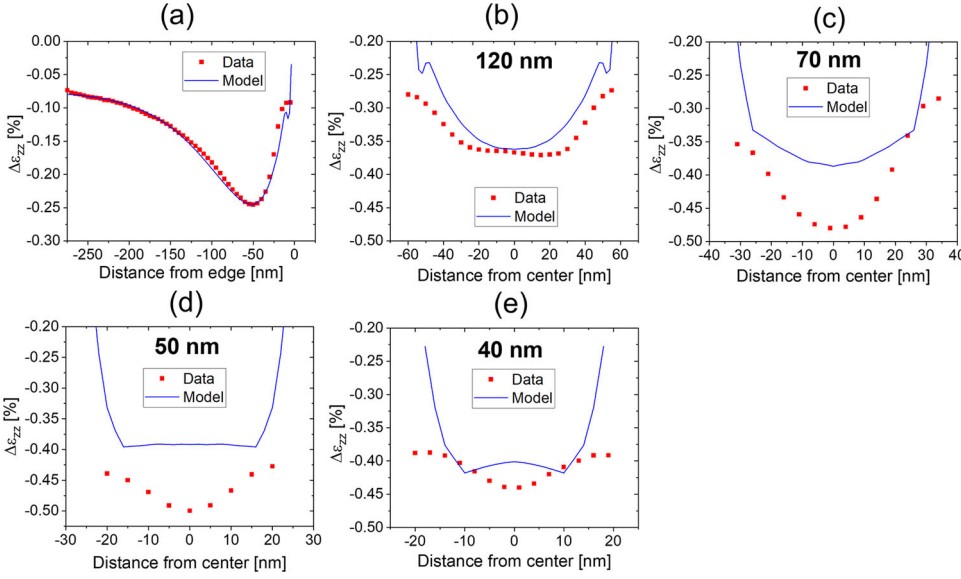

**Fig. 6 Depth-averaged Si/SiGe nanosheet deformation.** Comparison of the relative strain in the nanosheets (red squares) to finite element method (FEM) modeling (blue lines) as a function of width **a** blanket film edge, **b** 120 nm, **c** 70 nm, **d** 50 nm, and **e** 40 nm.

effects of fabrication were also employed to calculate the strain of the nanosheet structures. Figure 6 contains a comparison of the FEM modeling of the depth-averaged, SiGe deformation to the measured data collected on nanosheets with different widths. The experimental and simulated distributions of $\Delta\varepsilon_{zz}$ match well near the blanket nanosheet edge (Fig. 6a), confirming the magnitude of the deformation associated with the blanket SiGe layers of 2.45%. The correspondence between the measured and simulated SiGe relaxation starts to diverge for the 120 nm wide nanosheet (Fig. 6b) and becomes more pronounced as the nanosheet width decreases to 50 nm (Fig. 6d). This trend may be attributed to several factors such as averaging over the finite size of the x-ray probe, potential defectivity at the nanosheet sidewalls, or compositional fluctuations within the nanosheets. Because the first mechanism should lead to an apparent decrease of relative deformation rather than greater from the diffraction measurements, we do not believe that the convolution of the strain across the probe width is responsible. Similarly, plastic relaxation induced by defectivity would reduce the magnitude of deformation within the nanosheets. While the diffraction measurements indicate a small shift in the average effective Ge fraction within the SiGe nanosheets (36.3%) relative to the layers prior to patterning (36.7%), approximately 99% of the residual strain is retained in the nanosheets, and does not explain the discrepancy. SiGe deformation integrated over the entire 40 nm wide nanosheet relative to blanket SiGe (Fig. 6e), approximately −0.41%, is similar to the geometric phase analysis results reported in ref. [10] (2.0–2.45%). However, the detailed deformation distributions within the nanosheets reveal the impact of nanosheet width; the 120 nm wide nanosheets exhibit a central region (approximately 60 nm) with nearly constant deformation while the narrower nanosheets follow a more common profile that differs in magnitude. This behavior is a manifestation of how the deformation fields induced by the two opposing edges of the nanosheets interact, where the FEM modeling predicts less overall relaxation than observed in the diffraction data. While the characteristic features of the deformation field are captured by the mechanical simulations, future work will investigate the quantitative differences.

Precession electron diffraction measurements[10] of similar nanosheets revealed a similar discrepancy between measured and simulated elastic relaxation, which was due to the presence of an overlying silicon nitride cap[20]. However, FEM simulations conducted on SiGe nanosheets from this study using different elastic moduli of the capping layer displayed negligible changes in the resulting SiGe deformation, suggesting that the silicon nitride has a minimal influence. The synchrotron-based, diffraction measurements, which can be obtained without the uncertainty of elastic relaxation normal to the electron-transparent lamella, reveal how these competing mechanisms dictate the overall strain state within nanosheet builds without TEM sample preparation and at a spatial resolution commensurate with the deformation. This level of fidelity is essential in linking such strain distributions within the nanosheet to their electrical mobility, where the current-carrying regions are localized near their edges[43], and future implementations of these nanosheets will incorporate such strained layers in order to achieve the necessary performance gains necessary in future technology nodes[44].

## Conclusions

Nanoscale mapping of the deformation field in Si/SiGe nanosheet features using non-destructive x-ray diffraction, accomplished using a hard x-ray nanoprobe with a precision of strain up to $10^{-4}$ and spatial sensitivity on the order of several nanometers, revealed two length scales with respect to the deformation: a longer one related to elastic relaxation of the average, in-plane lattice mismatch and a finer one generated by the out-of-plane accommodation between the alternating Si and SiGe layers. As the nanosheet width decreases, the latter mechanism will dictate the deformation observed within the structures. This study sheds light on the mechanical response of complex nanostructures that can be visualized at extremely fine dimensions, as deviations in the observed strain distributions from those based solely on simulations can significantly influence the predicted mobilities and corresponding performance of current and future generation nanoelectronic devices.

## Methods

**X-ray nanodiffraction measurements**. Diffraction measurements were performed at the HXN beamline of the NSLS-II at the Brookhaven National Laboratory[32,34]. A pixel-array detector (Merlin, Quantum Detectors) with a 55-micron pixel size was used to record diffraction patterns at a downstream position of 400 mm, and

fluorescence photons were captured by an energy-dispersive detector (Vortex, Hitachi) placed at a direction orthogonal to the incident beam. While the use of an x-ray nanobeam provides a high spatial resolution, the corresponding analysis in reciprocal space contains additional complexity due to the stringent focusing conditions. As Fig. 2b shows, the measured RSM represents a convolution of the reciprocal functions associated with the sample, the beam, and the detector. With a 55-micron pixel size and a detector-to-sample distance of 400 mm, the streak associated with the finite detector pixel spans an angle of about 0.01 degrees. From Bragg's law, $\Delta d/d = -\Delta\theta/\tan(\theta)$, where $\Delta\theta$ is the change of the Bragg peak position and $\theta$ is the Bragg angle. For the Si (004) reflection, $\theta$ is 22.36° at a photon energy of 12 keV. Using sub-pixel interpolation, a strain sensitivity on the order of $1 \times 10^{-4}$ can be achieved. However, the nanobeam possesses an angular divergence of approximately 0.6 degrees, which also leads to a long streak, as seen in the RSM (Fig. 2c), and complicates the extraction of strain from the sample. Because the lengths of the nanosheets were approximately 100 microns, much larger than their width, scans conducted near the center of the nanosheets with respect to the x-direction would not exhibit lattice tilting in the $xz$ plane relative to the substrate.

**Mechanical modeling**. BEM modeling, based on ref. [23], using 2D geometry was conducted where a plane strain condition was assumed in the longitudinal direction of the nanosheets. An eigenstrain of 1.39%, corresponding to the lattice mismatch between the SiGe and the underlying Si substrate, was applied to the SiGe sheets so that the overall nanosheet geometry was allowed to deform in a linear, elastic manner. FEM modeling was conducted using Sentaurus Process™ (Synopsys, Inc., Mountain View, CA) which incorporated the effects of nanosheet fabrication on their residual strain state. Deformation profiles averaged through the thickness of each of the layers of the Si/SiGe stack as a function of lateral position were produced under the assumption of plane strain along the nanosheet length.

## Data availability
Data contained in this manuscript are available from the corresponding authors upon reasonable request.

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

## Acknowledgements

This research used 3ID beamlines of the National Synchrotron Light Source II, a US Department of Energy (DOE) Office of Science User Facility operated for the DOE Office of Science by Brookhaven National Laboratory under Contract No. DE-SC0012704.

## Author contributions

N. Loubet and C.L. prepared the samples; C.M., H.Y., C.L., J.J.S., and A.P. performed the x-ray measurements and K.R. the transmission electron microscopy; H.Y. did the data analysis; C.M., M.H., N. Lanzillo, and R.R. conducted the mechanical modeling; and C.M. and H.Y. wrote the manuscript.

## Competing interests

The authors declare no competing interests.
