## [Peer Review File · Communications Engineering]

Reviewers' comments:

Reviewer #1 (Remarks to the Author):

Review, Engineering Communications, Murray, Yan et al.

This manuscript describes the challenge of characterizing strain in nanoscale heterostructures like the technologically relevant SiGe/Si nanosheets studied in this work. By using state of the art optics (multi-layer-laue lenses) at HXN of NSLS-II they are able to push the limits for real-space X-ray imaging. This is a challenging experiment with such high resolution optics and a challenging and interesting sample. They conclude that the nanosheets show qualitatively similar behavior to BEM and FEM simulations, a long range deformation due to the sidewalls and strain between the Si/SiGe interlayers. This is supported by measurements of nanosheets of different thicknesses. One challenge within this work is that despite the high resolution optics, the spatial resolution is still not sufficient for imaging the interlayer strain in these structures. The authors correctly note the infeasibility of doing higher resolution X-ray imaging (ptychography) given the small volume of this sample. Instead, the authors used a dynamic diffraction model to fit the radial scans of the diffraction patterns. This works well for the Si blanket, but does not fit very well for the nanosheet stacks. The authors then allow the lattice parameter to vary between layers to achieve a better fit. I think this could be supported with more knowledge of the composition in the layers – perhaps the composition varies between layers which is why a single lattice parameter does not fit well.

The authors conclude that the BEM simulations and long range deformation of the nanosheets match well. However, the FEM does not quantitatively match the measurements of the strain in the interlayers, predicting less relaxation than actually observed. This is the more difficult imaging direction and would benefit from higher spatial resolution, but overall this finding still underpins the need for experimental strain mapping for nanoscale structures – we cannot only rely on mechanical simulations.

While I think this work could be strengthened a bit, overall it seems of high quality both in experimentation and analysis. I would recommend publication with minor revisions. Please see specific suggestions below.

1.

Figure 2 (c):

Quantitative labels of the q_x and q_z axes would help.

2.

Figure 3 (a/c):

This is a confusing figure, particularly (a) and (c). In this image I think that the x-axis is position along the sheet length... if so it would be helpful to label the axis (a, c). It is very difficult for me to understand what to take away from the q_z axis. Describing it as “stacked scans” doesn’t explain very well.

3.

Figure 3 (d):

The label says that Ge k-a fluo, out-of-plane lattice def and lattice rotation are plotted as a function of position. But only out-of-plane lattice def is plotted.

4

Pg 7 within "Data Analysis and Results":

Strain distributions between varying width nanosheets are imaged. The 120 nm wide sheets contain a central region that differs (slight decrease or flat), which is noted to be different than the behavior in the 70, 50, and 40 nm sheets. I think there should be more discussion about the limitations of spatial resolution on determining the shape of the strain profiles across the narrow sheets. Can we really differentiate the behavior of the 120nm sheet? Or is it simply the only one large enough to get a resolvable strain profile?

5.

Pg 8, paragraph 1:

How was the composition determined? From growth parameters or from actual measurements on similar samples, say with EDS? This is an important component of the analysis for determining the strain between interlayers.

Constant lattice spacing does not produce a good fit, so strain in each layer is allowed to vary. Could the composition be what is varying? I don't think that effect of composition variations is considered enough. Can these structures be imaged with EDS to see if there are fluctuations?

6.

Pg 10, paragraph 1:

The simulations in Figure 6 diverge for the smaller nanosheets. This is attributed to potentially 2 things: defectivity at the nanosheet sidewalls and averaging over the x-ray probe size. The probe-size effect is ruled out qualitatively, but some pretty simple kinematic scattering simulations (using the beam profile and the step size) could easily confirm this, and give a more direct comparison between the simulations and data for Figure 6.

7.

Given the conclusions, I think a more detailed discussion of how this strain distribution will modify device performance would strengthen the conclusions. Specifically why is knowing this strain profile going to inform future growth and how would this strain state help or hurt the device performance.

Reviewer #2 (Remarks to the Author):

The submitted manuscript has studied the measurement of the strain of Si / SiGe nanosheets designed for GAA FETs beyond 7nm node, and two distinct elastic relaxation mechanisms were directly observed in the nanosheets. Different characterized technology and modeling were carried out to confirm the results. The authors have demonstrated the mechanical response of Si/SiGe nanostructures. The results are important to the design and prediction of the strain-enhanced properties of current and future advanced nanodevices. However, the bellow questions still need to be noted for the authors:

1. In the experiment part, "Probes with a spot size of approximately 10 nm have been realized at several synchrotron facilities"? Here, 10 nm is the area or diameter of the spot? What is your sample size for synchrotron XRD measurement?
2. What is your growth method for Si/SiGe nanosheets? Which tool? The content is 36%, how to

control the mismatch and dislocation is very challenging. Please add some sentences about this content.

3. SiN is usually employed as a stress liner for strain engineering. Have you considered this part in your experiment or modeling? Do you try to use SiO or without SiN to compare the results?

4. Have you evaluate the defects' impactions in the deformation?

Reply to reviewers:

We would like to thank the reviewers for their constructive comments and suggestions. We have itemized our responses to each of them below:

Reviewer #1:

- 1) **Figure 2 (c): Quantitative labels of the q_x and q_z axes would help.**

Numerical labels have been added to the q_x and q_z axes in the revised version of Figure 2c.

- 2) **Figure 3 (a/c): This is a confusing figure, particularly (a) and (c). In this image I think that the x-axis is position along the sheet length... if so it would be helpful to label the axis (a, c). It is very difficult for me to understand what to take away from the q_z axis. Describing it as “stacked scans” doesn’t explain very well.**

We have revised this figure, its caption and the corresponding text on page 6 in hopes of removing any ambiguity. The x-axis does indeed refer to position along the nanosheet, and the z-axis refers to the out-of-plane, reciprocal space component, q_z . Because ‘stacked’ scans refers to an array of individual reciprocal space scans as a function of (real-space) position, we have changed the description of Figure 3a to ‘Contour plot of (004) diffracted intensity (on a logarithmic scale) as a function of q_z (vertical axis) and position (horizontal axis) across the edge ...’. Figure 3c represents the same type of information collected from nanosheets possessing different width, and its caption has been similarly modified.

- 3) **Figure 3 (d): The label says that Ge k-a fluo, out-of-plane lattice def and lattice rotation are plotted as a function of position. But only out-of-plane lattice def is plotted.**

The reviewer is absolutely correct – the erroneous figure caption resulted from a prior compaction of different figures. It has been corrected to reflect that only out-of-plane lattice deformation is displayed in Fig. 3d.

- 4) **Pg 7 within “Data Analysis and Results”: Strain distributions between varying width nanosheets are imaged. The 120 nm wide sheets contain a central region that differs (slight decrease or flat), which is noted to be different than the behavior in the 70, 50, and 40 nm sheets. I think there should be more discussion about the limitations of spatial resolution on determining the shape of the strain profiles across the narrow sheets. Can we really differentiate the behavior of the 120nm sheet? Or is it simply the only one large enough to get a resolvable strain profile?**

One point that we tried to make in the manuscript (on page 7) is that the observed deformation profiles in these nanosheets result from the interaction of the strain fields generated by the opposing nanosheet edges. This is where the importance of the single edge associated with the ‘blanket’ nanosheet (Figure 6a) is relevant, as it demonstrates an interaction region of at least 300 nm extending from the edge, for this nanosheet geometry. Thus, the central region in the 120 nm wide nanosheets does not represent a region absent of these interactions but rather the complex interplay between the two strain fields. For the narrower nanosheets, the overall profile lacks a ‘flat’ region because of this interplay, rather than a consequence of the resolution of the nanodiffraction experiments. In fact, FEM modeling shown in Figure 6a also shows that this interaction region extends well beyond the width of the nanosheets investigated here (120 nm and below) so that the differences among the simulated deformation profiles (Figure 6b to 6e) are also due to the nuances of the interplay between the edges. Because the measurements clearly differentiate the response of the 120 nm wide nanosheets from the ‘blanket’ edge nanosheet sample, it suggests we are sensitive to the deformation within these structures. We have added text to page 7 of the revised manuscript to address these points.

- 5) Pg 8, paragraph 1: How was the composition determined? From growth parameters or from actual measurements on similar samples, say with EDS? This is an important component of the analysis for determining the strain between interlayers. Constant lattice spacing does not produce a good fit, so strain in each layer is allowed to vary. Could the composition be what is varying? I don't think that effect of composition variations is considered enough. Can these structures be imaged with EDS to see if there are fluctuations?

The composition of the nanosheet layers was assessed prior to lithographic patterning using high-resolution XRD. Fitting of the radial scans about the (004) reflection was performed with RADSTM software to deduce the effective Ge fraction within each of the 3 SiGe layers, as is commonly undertaken for epitaxial layers deposited on single crystal substrates. This effective Ge fraction is a better metric for the purposes of calculating the deformation distributions than the absolute Ge concentration, as the former quantity corresponds to the substitutional Ge content that induces strain within the nanosheets. An effective Ge concentration of $36.7 \pm 0.4\%$ was inferred in the SiGe layers by this fitting under the assumption of pseudomorphic growth. Because the variation in Ge composition (0.4%) among the layers is less than 1.1%, it does not explain the discrepancies between the measured and modeled deformation in the nanosheets. Text has been added to page 4 of the manuscript to address this point.

To answer the question regarding compositional uniformity, we conducted EDS analyses of the individual SiGe layers within a 40 nm wide nanosheet structure. Below is a plot of the ratios of the integrated Si K counts to Ge L counts at each point across the nanosheet width, where the ‘0’ value of the x-axis corresponds to a position within 5 nm of the nanosheet center. Although one would need the appropriate calibration scheme to convert these ratios

into absolute Ge concentrations, they should be sufficient in allowing us to assess any variation across the nanosheets.

The dotted lines correspond to the average ratios within each of the SiGe layers: 0.97, 0.95 and 0.97 for the top, middle and bottom layers, respectively. We observe that all but 2 of the data points lie within one standard deviation, as determined from 3 separate measurements at each point, of these averages. However, it is interesting to note that the outermost points exhibit the lowest ratios (i.e: largest Ge fractions) in the middle and bottom layers, suggesting that heterogeneity in the Ge concentration may not be completely absent across the nanosheets but is not large enough to explain the difference between the measured and simulated deformation profiles. As such, we have modified the text on page 10 of the revised manuscript to suggest that composition fluctuations within the nanosheets could contribute to the observed discrepancies.

- 6) Pg 10, paragraph 1: The simulations in Figure 6 diverge for the smaller nanosheets. This is attributed to potentially 2 things: defectivity at the nanosheet sidewalls and averaging over the x-ray probe size. The probe-size effect is ruled out qualitatively, but some pretty simple kinematic scattering simulations (using the beam profile and the step size) could easily confirm this, and give a more direct comparison between the simulations and data for Figure 6.

We recognize the reviewer's comment that the finite size of the x-ray beam can lead to a broadening of the measured deformation distributions. The plots below (left: 50 nm wide nanosheet, right: 70 nm wide nanosheet) compare diffraction data ('HXN') to the simulated

deformation profiles ('FEM'), in which we also include FEM values convolved with an assumed Gaussian probe shape possessing FWHM values of either 18 nm or 40 nm.

While the shape of the FEM values with a 40 nm FWHM beam size (which we believe significantly overestimates the actual x-ray probe) may resemble the profiles of the x-ray diffraction data, the absolute values clearly do not match. As we mention in the text on page 10 of the manuscript, the corresponding decrease of relative deformation induced by beam convolution leads to a greater disparity between the simulated and measured values. Thus, we feel the cleanest assessment of the results in Figure 6 should incorporate the unconvoluted simulated values.

The reviewer also brings up a good point that defects within the heteroepitaxial layers can modulate the magnitude of strain within these types of structures, and motivated us to refine the assessment of the deformation within the nanosheets after their fabrication. Correlating the simulated deformation profile to the measured values for the blanket nanosheet edge (Fig. 6a), we find an out-of-plane deformation of 2.45%, corresponding to an effective Ge fraction of 36.3%. We have revised all of the simulated deformation profiles in Figure 6 using an effective Ge fraction of 36.3%. While this fraction lies within the error bars associated with x-ray diffraction measurements of the blanket films prior to patterning ($36.7 \pm 0.4\%$), it represents an approximate decrease in the driving force of deformation by 1.1% (approximately 99% of the residual strain within the blanket SiGe layers). As such, this value provides a bound to plastic relaxation in the nanosheets due to defectivity but clearly does not explain the observed discrepancies in the nanosheet deformation profiles. We have mentioned these points on page 10 of the revised manuscript.

We also note that certain attributes of the diffraction patterns, such as the Pendellosung (or interference) fringes commonly observed in pseudomorphically deposited, heteroepitaxial layers on single crystal substrates, would be eliminated in the presence of substantial defectivity. Because these features are still present in the data we acquired from the

nanosheets (see Fig. 3a and 3c), we suspect that any defectivity is too low to cause the observed deviation between simulated and measured deformation profiles. A final point is that such defectivity is expected to lower the overall deformation within the nanosheets by plastic relaxation, similar to the effects of convolution due to beam size, which is counter to the larger deformation we observe as compared to the simulations.

- 7) Given the conclusions, I think a more detailed discussion of how this strain distribution will modify device performance would strengthen the conclusions. Specifically why is knowing this strain profile going to inform future growth and how would this strain state help or hurt the device performance.

As we described in the introduction, distributions of the strain state within the current-carrying regions of the nanosheets will induce different carrier mobilities and thus will directly impact their electrical response. References 13 to 15 in the original manuscript discuss how the piezoresistive behavior of semiconductor materials (Si and Ge) dictate this response due to deformation. The clear challenge, and reason for experimental validation of the actual strain within the nanosheets, is that electrical measurements only produce an aggregate value of this response; reconstructing the unique deformation solely from electrical measurements is an intractable task particularly when variations exist within the devices, as described in Ref. 7 in the manuscript. We also tried to emphasize this point in the paragraph preceding the Conclusions, where we stated that it is essential to link such strain distributions within the nanosheet to its electronic mobility, and future implementations of these nanosheets will incorporate such strained layers in order to achieve the necessary performance gains necessary in future technology nodes. To accommodate the reviewer's request, we have provided additional text at the end of the Conclusions section on page 12 of the revised manuscript.

Reviewer #2:

- 1) In the experiment part, "Probes with a spot size of approximately 10 nm have been realized at several synchrotron facilities"? Here, 10 nm is the area or diameter of the spot? What is your sample size for synchrotron XRD measurement?

The spot size mentioned in the manuscript corresponds to the diameter of a focused x-ray beam in normal incidence, as does the 12 nm value we provided on page 3 of the original manuscript. The in-plane dimensions of the sample we analyzed are approximately 3 mm x 3 mm, and approximately 0.75 mm thick corresponding to the underlying Si substrate.

- 2) What is your growth method for Si/SiGe nanosheets? Which tool? The content is 36%, how to control the mismatch and dislocation is very challenging. Please add some sentences about this content.

The epitaxial growth of the SiGe layers was performed on 300 mm diameter, (001)-oriented silicon wafers in a commercially available, rapid-thermal chemical vapor deposition reactor. Although we cannot release all of the information related to processing of the SiGe nanosheets, Reference 8 of the original manuscript details the aspects of depositing epitaxial SiGe films while minimizing defectivity that can lead to plastic relaxation. As mentioned in our response to the first reviewer's fifth comment, fitting of x-ray diffraction data collected on the blanket layers prior to nanosheet fabrication allowed us to deduce the effective Ge fraction within each of the 3 SiGe layers, as is commonly undertaken for epitaxial layers deposited on single crystal substrates. This effective Ge fraction is a better metric for the purposes of calculating the deformation distributions than the absolute Ge concentration, as the former quantity corresponds to the substitutional Ge content that induces strain within the nanosheets. An effective Ge concentration of $36.7 \pm 0.4\%$ was inferred in the SiGe layers by this fitting under the assumption of pseudomorphic growth. Text has been added to page 4 of the manuscript to address these points.

- 3) SiN is usually employed as a stress liner for strain engineering. Have you considered this part in your experiment or modeling? Do you try to use SiO or without SiN to compare the results?

Yes - because we had similar concerns, we conducted simulations of the potential impact of the overlying SiN hardmask on the strain distributions within the underlying nanosheets. In particular, we reduced the elastic modulus, E_{SiN} , of the SiN structures by a factor of 2, which exhibited less than a 2% difference in the calculated deformation as shown below for the 120 nm wide nanosheet geomtry:

Thus, we conclude that the SiN layer has a minimal influence on the strain state within the nanosheets, as stated on page 11 of the revised manuscript.

4) **Have you evaluate the defects' impactions in the deformation?**

This point is similar to the first reviewer's sixth comment. Correlating the simulated deformation profile to the measured values for the blanket nanosheet edge (Fig. 6a), we find an out-of-plane deformation of 2.45%, corresponding to an effective Ge fraction of 36.3%. We have revised the simulated deformation profiles in Figure 6 using an effective Ge fraction of 36.3%. While this fraction lies within the error bars associated with x-ray diffraction measurements of the blanket films prior to patterning ($36.7 \pm 0.4\%$), it represents an approximate decrease in the driving force of deformation by 1.1% (approximately 99% of the residual strain within the blanket SiGe layers). This value provides a bound to plastic relaxation in the nanosheets due to defectivity, which we have added to page 10 of the revised manuscript.

We also note that certain attributes of the diffraction patterns, such as the Pendellosung (or interference) fringes commonly observed in pseudomorphically deposited, heteroepitaxial layers on single crystal substrates, would be eliminated in the presence of substantial defectivity. Because these features are still present in the data we acquired from the nanosheets (see Fig. 3a and 3c), we suspect that any defectivity is too low to cause the observed deviation between simulated and measured deformation profiles. And such defectivity is expected to lower the overall deformation within the nanosheets by plastic relaxation, which is counter to the larger deformation we observe as compared to the simulations.

REVIEWERS' COMMENTS:

Reviewer #1 (Remarks to the Author):

Authors have done a good job addressing both reviewers comments and questions. I would recommend this work for publication without any further revisions.

Reviewer #2 (Remarks to the Author):

I have no further comments. I think the revised version should be accepted.